# Model-Based Integration Analysis Revealed Presence of Novel Prognostic miRNA Targets and Important Cancer Driver Genes in Triple-Negative Breast Cancers

**DOI:** 10.3390/cancers12030632

**Published:** 2020-03-09

**Authors:** Masood Zaka, Chris W. Sutton, Yonghong Peng, Savas Konur

**Affiliations:** 1School of Health & Life Sciences, Teesside University, Middlesbrough TS1 3BA, UK; M.Zaka@tees.ac.uk; 2National Horizon Centre, Teesside University, 38 John Dixon Lane, Darlington DL1 1HG, UK; 3Institute of Cancer Therapeutics, School of Life Sciences, University of Bradford, Bradford BD7 1DP, UK; C.W.Sutton@bradford.ac.uk; 4Department of Computing and Mathematics, Manchester Metropolitan University, Manchester M15 6BH, UK; 5Department of Computer Science, Faculty of Engineering and Informatics, University of Bradford, Bradford BD7 1DP, UK

**Keywords:** genomics, epigenetics, microRNA, integrated analysis, triple negative breast cancer, biomarkers and signalling pathway

## Abstract

*Background*: miRNAs (microRNAs) play a key role in triple-negative breast cancer (TNBC) progression, and its heterogeneity at the expression, pathological and clinical levels. Stratification of breast cancer subtypes on the basis of genomics and transcriptomics profiling, along with the known biomarkers’ receptor status, has revealed the existence of subgroups known to have diverse clinical outcomes. Recently, several studies have analysed expression profiles of matched mRNA and miRNA to investigate the underlying heterogeneity of TNBC and the potential role of miRNA as a biomarker within cancers. However, the miRNA-mRNA regulatory network within TNBC has yet to be understood. *Results and Findings*: We performed model-based integrated analysis of miRNA and mRNA expression profiles on breast cancer, primarily focusing on triple-negative, to identify subtype-specific signatures involved in oncogenic pathways and their potential role in patient survival outcome. Using univariate and multivariate Cox analysis, we identified 25 unique miRNAs associated with the prognosis of overall survival (OS) and distant metastases-free survival (DMFS) with “risky” and “protective” outcomes. The association of these prognostic miRNAs with subtype-specific mRNA genes was established to investigate their potential regulatory role in the canonical pathways using anti-correlation analysis. The analysis showed that miRNAs contribute to the positive regulation of known breast cancer driver genes as well as the activation of respective oncogenic pathway during disease formation. Further analysis on the “risk associated” miRNAs group revealed significant regulation of critical pathways such as cell growth, voltage-gated ion channel function, ion transport and cell-to-cell signalling. *Conclusion*: The study findings provide new insights into the potential role of miRNAs in TNBC disease progression through the activation of key oncogenic pathways. The results showed previously unreported subtype-specific prognostic miRNAs associated with clinical outcome that may be used for further clinical evaluation.

## 1. Introduction

Triple-negative breast cancer (TNBC) is a heterogeneous group of cancers defined by lack of expression of estrogen receptor (Er), progesterone receptor (Pr) and human epidermal growth factor receptor2 (Her2) [1,2]. The triple-negative subgroup accounts for 15% of all types of breast cancer, representing an aggressive disease with limited treatment options and inadequate molecular prognostics signatures. Recent advances in the Omics technologies (genomics, epigenomics and proteomics) have improved our understanding of molecular complexities and identification of novel biomarkers for TNBC [3]. However, despite unprecedented progress in technology and disease understanding, clinical decisions are still based on the assessment of three receptors.

miRNAs (MicroRNAs) are a family of small (~19–25 nucleotides), noncoding RNAs that reduce the abundance and translational efficacy of mRNAs during the transcription process [4,5]. Determining the role of individual miRNAs is a major challenge due to the biology of miRNA, complexity of cancer disease and unknown cellular regulatory functions. miRNAs are very well characterised into families but only a few known miRNAs have been associated with functionally characterised tissue types [6]. 

Extensive investigation of mRNA expression profiles has been undertaken for the identification of diagnostic markers for control and disease [7,8,9,10]. Similarly, in recent years multiple studies have been conducted, exploiting the expression profiles of miRNA and mRNA for the discovery of new specific miRNA signatures, and candidate transcription factors [11]. Studies discussing the characterisation of miRNAs have confirmed the existence of complex networks targeting genes involved in critical functional processes such as differentiation, cell cycle and apoptosis [12,13,14].

The number of miRNAs characterised and deposited to the miRBase database has seen an exponential rise since the discovery of the first miRNA [4,15,16,17,18]. Therefore, the identification of significant miRNAs and their target genes is of great importance for the understanding of complex diseases such as TNBC. It has also been observed that comprehensive functional annotation miRNAs and their targets still remain a major challenge despite the development of multiple high-throughput methods [19]. Several computational algorithms and methods such as PITA [20], TargetScan [21] and miRwalk [21] have been developed recently to address this problem [22], but sequence-similarity-based gene targeting still remains the widely used method for the annotation of miRNAs. 

In this study, we have performed a model-based integrated analysis using the expression profiles of miRNAs and mRNA samples extracted from triple-negative breast cancers. The analysis has showed that the triple-negative subtype can be distinguished from other subtypes of breast cancers using miRNA expression profiling techniques. The study has also revealed that subtype-specific miRNAs are potential biomarkers for DMFS and OS, and are strongly anti-correlated with known cancer driver genes of oncogenic pathways. Our findings reflect the existence of heterogeneity among triple-negatives and also provide evidence for miRNA influence on tumour transcriptional phenotypes by targeting coding genes. Our results show that molecular studies based on miRNAs biomarkers can help in early detection and novel therapeutic targets for triple-negatives.

## 2. Results

### 2.1. miRNAs Differentially Expressed among the Tumour Subtypes 

The expression profiles of 142 breast tumour samples with known biomarker statuses of Er, Pr and Her2 receptors were considered for the identification of differentially expressed miRNAs. The subgroup comparison of 94 TNBC, 40 Non-TNBC, and eight normal showed the presence of 206 differential miRNAs. A total of 172 miRNAs were significantly dysregulated in triple-negative versus non triple-negative comparison, mostly with known involvement in breast cancer [23,24]. miRNAs exhibiting *p*-value < 0.05 were considered statistically significant and processed for downstream analysis. 

Unsupervised hierarchical clustering analysis has separated 172 dysregulated miRNA genes into three major clusters based on expression profiles and sample information from TNBC and non-TNBC patients (Figure 1C). 

Cluster 1: This cluster comprises 53 significantly down-regulated miRNAs in TNBC tissues. Those with a two-fold change are hsa-miR-92a (logFC −1.08); hsa-miR-19a (logFC −1.06); hsa-miR-17 (logFC −1.05); hsa-miR-135b (logFC −1.4); hsa-miR-20a (logFC −1.07); hsa-miR-20a* (logFC −1.07) and hsa-miR-9* (logFC −1.08), and all are involved in important KEGG pathways such as pathways in cancer, colorectal pathways, bladder cancer and TFG-beta signaling pathway by targeting genes like *NRAS, APC, RUNX1, BCL2, MAPK9, CCND1, SMAD4, HIF1A, MYC, CDKN1A, VEGFA, PTEN, TGFBR2*, and *JAK1* (Appendix A).

Cluster 2: This cluster consists of 24 down-regulated and a subgroup of three up-regulated miRNAs among the TNBC patients (hsa-miR-936, logFC 0.06; hsa-miR-622, logFC 0.11; hsa-miR-492, 0.05). We observed down-regulation of hsa-mir-574-5p (logFC −0.29) and hsa-mir-320c (logFC −0.36), which targets *RAB18* gene, a member of *RAS* oncogene family [25], *ABHD13* [26], *DKDG* [27] and *FOX12* [28]. hsa-mir-320c also indirectly regulates TNBC by targeting known biomarker genes such as *MCL1* [29] and *RAD51* [30]. Another established breast cancer prognostic miRNA, hsa-mir-1290, was down-regulated (logFC −0.58) TNCB samples and target critical genes such as *SGOL1*, a member of Shougoshin family member [31], *FAM53C* [32] and transcription factor II-I *GTF21* [33]. 

Cluster 3: Most members of this clusters were strongly up-regulated, except a subgroup of five miRNAs (hsa-miR-155, logFC −0.35; hsa-miR-146b-5p, logFC −0.30; hsa-miR-146a logFC −0.66; hsa-miR-31*, logFC −0.48, hsa-miR-31 logFC −0.48) which were under-expressed and known to be involved in oncogenic activities by positively regulating sodium–iodide symporter (NIS) expression. As previously reported, up-regulated hsa-mir-127-3p in breast-cancer-negative tumours has been involved in inhibiting growth, enhanced apoptosis and migration in breast cancer cells by targeting BCL-6 oncogene [34], whilst hsa-miR-654-3p (logFC 0.07) regulates proliferation, apoptosis, migration and invasion in prostate cancer [35]. Two members of the mir-342 family were up-regulated, hsa-mir-342-3p (logFC 1.10) and hsa-mir-342-5p (logFC 0.48), in profiles and are known to be involved in the regulation of BRCA1 expression [36]. Other miRNAs with oncogenic activities from this cluster were significantly up-regulated, such as hsa-miR-375 (logFC 1.58), and could potentially act as diagnostic/prognostic biomarkers [37,38]. Similarly, two members of mir-92-c family mediate epithelial-to-mesenchymal transition in human cancers by regulating β-catenin signalling [39]. Two over-expressed miRNAs belong to the family of hsa-mir-193 (hsa-mir-193b*, logFC 0.70 and hsa-mir-193a-5p, logFC 0.36) and two individual miRNAs, hsa-mir-665 (logFC 0.08) and hsa-miR-663b (logFC 0.04), were also increased in TNBC compared to non-TNBC.

### 2.2. miRNAs Associated with Survival 

Univariate cox regression on 172 differentially expressed miRNAs between triple negative versus non-triple negative breast tumours confirmed a set of 28 miRNAs associated with clinical outcomes. Three miRNAs (hsa-miR-342-3p/5p and hsa-miR-497) among the 28 miRNAs showed significant association with both DMFS and OS outcome that may indicate their ability to classify the patient with good and bad prognoses. A subset of 18 miRNAs were significantly associated with DMFS, whereas another group of 10 miRNAs showed a strong association with overall survival outcome. Of the former subset of miRNAs, nine were identified as up-regulated in the TNBC compared to non-TNBC (Table 1) patients. Interestingly, most of the up-regulated miRNAs, such as hsa-miR-29c (HR = 0.72, CI = 0.52–1), hsa-miR-342-3p (HR = 0.52, CI = 0.31–0.89), hsa-miR-342-5p (HR = 0.3, CI = 0.1–0.93), hsa-let-7c (HR = 0.63, CI = 0.41–0.98), hsa-let-7b (HR = 0.5, CI = 0.31–0.83), hsa-miR-369-5p (HR = 0, CI = 0–0.42), hsa-miR-101 (HR = 0.58, CI = 0.33–1), hsa-miR-497 (HR = 0.51, CI = 0.29–0.9) and hsa-miR-154 (HR = 0.05, CI = 0–0.58), were correlated with better prognosis. However, among the down-regulated miRNAs, three out of nine, including hsa-miR-1290 (HR = 1.71, CI = 1.2–2.43), hsa-miR-630 (HR = 1.64, CI = 1.17–2.3) and hsa-miR-1246 (HR = 1.53, CI = 1.12–2.09), were correlated with worse prognosis (risk-associated), and the remaining six, comprising of hsa-miR-19b-1 * (HR = 0, CI = 0–0.69), hsa-miR-301b (HR = 5.31, CI = 1.13–24.96), hsa-miR-181d (HR = 0.31, CI = 0.1–0.95), hsa-miR-181c * (HR = 0.1, CI = 0.01–0.76), hsa-miR-30e (HR = 0.49, CI = 0.25–0.98) and hsa-miR-130a (HR = 0.5, CI = 0.33–0.78), were linked with better prognosis, which indicates a dual role of gene regulation and inhibition of these miRNAs and heterogeneity within this subtype. 

The 134 breast tumour samples were divided into three groups (low, intermediate and high) based on their expression values (see material and methods for more detail) to investigate the robustness of these miRNAs as signature for survival outcomes. The 18 miRNA DMFS signature analysis indicated that tumours were strongly associated with intermediate risk (*N* = 44, HR = 1.24, CI = 1.04–1.47, *p*-value = 0.031) and high risk (*N* = 46, HR = 1.02, CI = 0.86-1.22, *p*-value = 0.031) compared to low-expressing tumours (Figure 2A, Table 1). Further investigation of the low risk samples suggested 40 non-TNBC patients and a group of four TNBC samples with similar expression profiles to non-TNBC (Figure 1C).

Similarly, 10 miRNAs were identified as significantly associated with overall survival (Figure 2B, Table 2). Three of these have also been observed to be strongly linked with good prognosis *p*-value < 0.05. The results from differential expression showed the up-regulation of seven miRNAs and down-regulation of three miRNAs within this group. Cox proportional hazard analysis of six up-regulated miRNAs including hsa-miR-342-3p (HR = 0.68, CI = 0.5–0.92), hsa-miR-342-5p (HR = 0.39, CI = 0.2–0.75), hsa-miR-195 (HR = 0.76, CI = 0.59–0.98), hsa-miR-936 (HR = 5.79, CI = 1.04–32.08), hsa-miR-338-3p (HR = 0.43, CI = 0.19–0.96), and hsa-miR-497 (HR = 0.64, CI = 0.44–0.94) were associated with good prognosis. Two of the down-regulated miRNAs within this group, hsa-miR-155 (HR = 0.61, CI = 0.41–0.91) and hsa-miR-146b-5p (HR = 0.65, CI = 0.45–0.94), were observed with low risk. The two remaining down-regulated miRNAs, hsa-miR-193b (HR = 1.5, CI = 1–2.25) and hsa-miR-1208 (HR = 376.22, CI = 10.32–13709.16), were significantly correlated with worse overall survival. miRNA hsa-miR-1208 was observed to be an outlier with an extreme increase in high hazard ratios and was excluded from downstream analysis. 

The 10-miRNA OS signature was further tested for its ability to distinguish the groups of (Low, Intermediate and High) expressing samples. The OS signature also split the 134 breast tumours into the three groups but with slightly less significance (*N* = 46, HR = 1.28, CI = 1.05–1.56, *p*-value = 0.04) and intermediate significance (*N* = 44, HR = 1.11, CI = 0.91–1.36, *p*-value = 0.04). A similar pattern of sample classification was observed as for the 18-miRNA DMFS signature. Surprisingly, a trend of down-regulated miRNAs’ association with high-risk was observed in both signatures including the outlier hsa-miR-1208 miRNA from the 10-miRNA OS signature. Similarly, the up-regulatory miRNAs from both signatures exhibited a strong association with low-risk, except hsa-miR-193b of 10-miRNA OS signature. 

We further analysed prognostic factors by conducting univariate and multivariate analyses using histopathological information on its own (without the miRNAs’ expression data). We observed node positivity and tumour size histopathological factors were significantly associated with DMFS prognosis (Appendix A), whereas the percentage of tumour cell infiltration, node positivity and tumour size were among the significantly correlated prognostic factors with OS (Appendix A). We then assessed the quality of the fitted model using analysis of deviance likelihood for the selection of co-variates which could impact on the association of prognostic factors with miRNA expression on the outcome prediction. The results from deviance-score analysis suggested that the two-factors (node positivity and tumour size) model was the best-suited model for DMFS prognosis, similar to clinical factors such as tumour size and percentage of tumour cell infiltration model for overall survival. Almost all miRNAs retained their prognostic ability when evaluated in multivariate models using information on node positivity, tumour size and tumour cell infiltration for DMFS and OS (Appendix A). 

### 2.3. In silico Validation 

In silico analysis showed that the proposed miRNA signature with 68% overlapping miRNAs with the Buffa et al. dataset shows significant association (HR = 1, CI = 1.00–1.00, *p*-value = 2.6E-05) with distant relapse-free survival (DRFS), whereas the miRNA signature showed a broad lack of reproducibility (*p*-value = 0.13) with over 80% overlap with the Bockhorn dataset when assessed for overall survival (Appendix A). This might be due to the sample scarcity and difference in the platforms or methods used during the samples collection. The heterogeneity within TNBC could also have played an important role in the performance of miRNAs in their prognostic ability.

### 2.4. mRNA Differentially Expressed among the Tumour Subtype Classes

The profiles of 18 triple-negative tumour samples and 131 non-TNBCs were used for differential expression analysis. The results showed that 386 genes were significantly expressed in TNBC tumours. Unsupervised hierarchical clustering separates 386 mRNAs into two large clusters with unique gene subsets (Appendix A). 

Cluster 1: This was a large cluster characterised by 180 down-regulated mRNAs. Of these, 19 mRNAs were among known biomarkers observed in previous studies linked to receptor-negative breast tumours. Validated biomarker genes included CDKN2A (logFC −2.44), involved in the G1/S checkpoint of the mitotic cell cycle and Ras protein signal transduction [40,41], EGFR (logFC −2.86) in the Fc-epsilon receptor signaling pathway, the activation of phospholipase C activity, the epidermal growth factor receptor signaling and cell proliferation [42,43], and IGF2BP3 (logFC −2.46) in mRNA transport and the regulation of the cytokines and biosynthesis process [44]. The canonical pathways observed for this cluster were cyclins and cell cycle regulation, cell cycle G1/S checkpoint, glioma, p53 signaling pathway, melanoma and Wnt signaling pathway enriched with EGFR, CCNE1, FZD9, CDKN2A, SFRP1, CDK6, SERPINB5, FGF9, MMP7, CALML5, TCF7L1, SHC4, and PRKX genes.

Cluster 2 included 206 mRNA genes significantly over-expressed in triple-negatives. The genes representative of this cluster were XBP1 (logFC 2.30) [45], SCGB2A2 (logFC 2.80) [46] and TFF3 (logFC 6.33), which have been previously associated with triple-negative tumours [47]. Other genes, such as ESR1 (logFC 6.02) [48]; GATA3 (logFC 3.88) [49]; ERBB4 (logFC 3.0) [50]; AR (logFC 2.94) [51]; CCDC170 (logFC 2.79) [52]; MAPT (logFC 2.45) [53]; PGR (logFC 2.91) [54]; PIP (logFC 5.42) in immunoglobulin G-binding protein [55] from this cluster were closely associated with breast cancer progression. The enrichment analysis for this cluster has identified the top perturbed biological processes, including response to estrogen stimulus involving genes BMP4, GSTM3, GATA3, ESR1, TFF1, NPY1R, and IGFBP2; cell differentiation (BMP4, AGTR1, METRN, IL6ST, MAPT, FOXA1, KITLG, and IL20); positive regulation of signal transduction (BMP4, HPX, IL6ST, ESR1, GJA1, F7, ECM1, IRS1, and IL20). In addition, other biological processes that were significantly affected include the extracellular matrix, voltage-gated channel activity, regulation of growth, regulation of cell motion and anti-apoptosis, cell adhesion, and cytokine binding. The top regulated pathways for this cluster were also linked with the downregulation of MTA-3, ERBB2 signal transduction and other oncogenic pathways. 

### 2.5. Prognostic miRNAs and Their Association with Predicted Targets and Enriched Functions

Target prediction analysis on differentially expression genes from the mRNA expression profiles identifies 35 mRNA genes with known involvement in various cancers. Of 28 miRNAs, 16 from both signatures are displayed in Figure 3 (Appendix A).

### 2.6. 18.-miRNAs Signature 

We further assessed the relationship of predicted miRNAs with each gene cluster of mRNA data set. Three low-risk miRNAs with significant low expression including hsa-miR-301b, hsa-miR-130a, hsa-miR-181d and one high-risk hsa-miR-1290 miRNA belong to the 18-miRNA DMFS prognostic signature targets cluster, two genes of the mRNA dataset involving REEP1 (logFC 2.16), CNTNAP2 (logFC 2.03), BMP4 (logFC 2.66), FOXA1 (4.33) and CLEC3A (logFC 3.61). These genes were functionally involved in processes such as neuron differentiation and the positive regulation of signal transduction. In general, all of the low-risk miRNA target genes were involved in important regulatory pathways of TNBC disease. hsa-miR-181d targets CACNA2D2 (logFC 2.34) gene, involved in important functions such as voltage-gated ion channel and ion transport. Other miRNAs such as hsa-miR-301b have shown anti-correlation with MUM1L1 (logFC 2.18) and NBEA (logFC 2.25), a regulator of plasma membrane. Highly up-regulated genes such as AFF3 (logFC 4.33) and GJA1 (2.02), targeted by another low-risk miRNA, hsa-miR-30e, are involved in KEGG pathways of arrhythmogenic right ventricular cardiomyopathy (ARVC) and other cellular processes like cell–cell signaling, endoplasmic reticulum and plasma membrane. 

Low-risk hsa-miR-29c miRNA targeted three downregulated mRNA genes, BCL11A (logFC −3.26), COL22A1 (logFC −3.56), involved in the regulation of important processes such as structural molecular activity and extracellular region; CDK6 (logFC −2.09), involved in multiple pathways of cancer, p53 signaling pathway, cell cycle regulation, and the positive and negative regulation of cell proliferation. Another low-risk hsa-miR-497 miRNA showed a strong anti-correlation with OTX1 (logFC −2.54), SH2D2A (logFC −2.04) and TRIM2 (logFC −2.24), all of which have demonstrated involvement in key cell functions (Appendix A). hsa-miR-101 was also a low-risk marker, targeting SOX11 (logFC −2.63) gene, a regulator of cell proliferation and glial cell differentiation and GABBR2 (logFC −2.63), involved in cell junction and cell projection molecular processes. hsa-miR-342-3p was negatively correlated with downregulated gene targets such as ZNF462 and NCOA7, and hsa-let-7b/c were anti-correlated with predicted targets KRT5 and GABBR2, with known activities in the cancer pathways.

### 2.7. 10.-miRNAs Signature 

Low-risk hsa-miR-155 miRNA targets genes like NOVA1 (logFC 2.14) and DACH1 (logFC 4.0), which have been observed to be involved in cell-to-cell signalling processes. Another low-risk miRNA hsa-miR-936 from this signature targets down-regulatory genes of mRNA cluster 2 such as IGF2BP2 (logFC −2.06) PTX3 (logFC −2.0) POU4F1 (logFC −2.83) that were functionally involved in alternative splicing, DNA-binding, mRNA transportation and signalling.

## 3. Discussion

Triple-negative disease is still characterised by the absence of three known receptor genes, Er, Pr, and Her2, because of underlying heterogeneity and aggressive disease behaviours. These two persistent challenges have slowed the progress of developing new targeted therapies within the triple negative subtype. Therefore, there is a need to identify novel biomarkers that can further stratify triple-negative into simpler molecular subtypes and may help in developing new screening techniques, the early detection of cancer, and better clinical outcome. The focus has recently been on elucidating novel and important elements, factors and mechanisms controlling transcription regulation. Investigating miRNAs involving and controlling transcriptional regulation may provide new understanding of complex diseases such as TNBC. Therefore, this study integrates information from miRNA and mRNA by developing independent models on genome-wide expression profiling and identifies important regulatory pathways, targetable markers for chemotherapy and their involvement in predicting clinical outcomes [5,56].

We investigated the role of miRNAs as a differentiating factor among the tumour subtypes based on expression profiling. We observed 173 miRNAs that clearly separate the breast tumours from their respective normal patients. Further analysis of the patients between triple negative and non-triple negative shows the presence of 172 miRNAs, dysregulated and targeting well-known biomarker genes within this subtype. These biomarker genes are involved in key pathways of signalling pathways such as the *p53* signaling pathway, *Ras* protein signalling and *Wnt* signalling pathway. Therefore, this offers strong a indication that these miRNAs indirectly control critical pathways and consequently impact the biology of TNBC microenvironments. Interestingly, the 77 common miRNAs between the triple negative and non-triple negative cases have shown higher expression in triple negative patients. The finding that miRNAs have higher expression in aggressive subtypes was previously confirmed by an independent study of Blenkiron et al. [57]. De Rinaldis et al., in their study, also showed that higher expression levels can be traced back to changes in the DNA copy numbers of these miRNAs [58]. Most of these highly expressed miRNAs originated from amplified regions of chromosomal DNA in the tumours. Further investigation into tumour progression from normal to tumour subtypes have indicated only three miRNAs (hsa-miR-154, hsa-miR-145, and hsa-miR-93) that are differentially expressed across the tissues (Figure 1B). The Venn diagram intersection between the three contrast tissue comparisons—normal versus non-TNBC, normal versus TNBC and TNBC versus non-TNBC—shows that the highest expression changed with triple-negatives. miRNAs hsa-miR-154 and hsa-miR-145 exhibited over-expression in TNBC compared to non-TNBC, whereas hsa-miR-93 is up-regulated when TNBC is compared with normal-like tissue. 

A large number of miRNAs from cluster 3 (Figure 1C) have shown higher expression, except the group of five miRNAs. miRNAs such as hsa-miR-342-3p, hsa-miR-195, hsa-miR-497, hsa-miR-29c, hsa-let-7c, and hsa-miR-497 were observed with higher expression in triple negatives and most of these were targeting genes from cluster 1 of mRNA expression data, which are highly down-regulated in the triple-negative patients. Most of these targets were known cancer driver genes involved in inhibiting tumour growth, speeding up the apoptosis, invasion and regulating migration in tumour cells (Figure 3, Appendix A). These miRNAs provide evidence of their acting as tumour suppressor in triple-negative and can be used as targets in tackling tumour progression and management; however, further validation will be needed. Another characteristic of these miRNAs is the formation of clusters in various cancers and their acting as modulators for important cell functions. It has been noted that these clusters usually reside very near to the known cancerous sites or genomics hotspots and play a very important role in carcinogenesis [59,60]. We identified two members of the miRNA *let* family hsa-let-7c/b targeting *KRT5* and *GABBR2*, involved in molecular functions such as G-protein-coupled signaling, protein binding and structural activity. The manipulation of this cluster can be used for experimental purposes and also for therapeutic intervention in triple-negative diseases. 

Next, we investigated miRNAs’ role as potential biomarkers in receptor-negative breast tumours and triple-negative disease. Using univariate Cox regression analysis, we identified 28 miRNAs significantly associated with survival outcomes that were further validated by applying multivariate analysis for their ability as independent markers. miRNA hsa-mir-1290 is down-regulated in triple-negative patients and has shown a strong association with higher risk groups with the triple-negatives. Previously, the over-expression of hsa-mir-1290 in the Her2-postive group has been reported and linked with better overall survival, as well as serving as a marker for early detection in triple-negatives [56,61]. Our results also showed strong evidence of hsa-mir-1290 as prognostic marker in triple-negative and its targeting two highly over-expressed genes (*BMP4*, logFC 2.66; *FOXA1*, logFC 4.33) in the mRNA dataset. *BMP4* and *FOXA1* have been reportedly involved in patients with relapse-free survival (RFS) and disease-free survival (DFS) in breast cancer [62,63]. Functionally, *FOXA1* is a member of the transcription family and a very well-characterised biomarker for triple-negatives and other carcinomas [51]. During the Cox regression analysis, we observed that almost all of the down-regulated miRNAs were associated with high-risk, and targeting up-regulatory genes in the mRNA dataset involved in cellular processes such as proliferation, angiogenesis and apoptosis. 

An emerging theme in the survival analysis was the association of up-regulated miRNAs with low-risk in triple negatives. miRNAs such as hsa-miR-29c target important biomarker genes like *CDK6*. This gene is involved in cell cycle checkpoints and regulation, the regulation of cell proliferation and the p53 signalling pathway. A recent paper was also in agreement with our finding that the down-regulation of hsa-miR-29c is linked to poor survival outcome in breast cancer patients [64]. Other predicted targets for hsa-miR-29c are BCL11A and COL22A1, involved in tumour formation, regulator of structural molecular activity and extracellular region. The low-risk miRNAs from has-let-7 family (hsa-let-7c, hsa-let-7b) act as tumour suppressors through controlling cellular processes such as cell projection, cell proliferation, and cell development, involving the genes KRT5 and GABBR2. Over-expression of hsa-miR-342-3p was a prognostic marker for DMFS and OS, and consistent with a better prognosis in our study. It has been reported that hsa-miR-342-3p regulates *BRCA1* expression and MYC transcriptional activity by directly repressing *E2F1* in human lung cancers [36,65]. Another low-risk miRNA hsa-miR-497 predicting DMFS and OS regulates several important signalling pathways related to VEGF, ErbB1 and stem cells pluripotency, involving gene targets SH2D2A, OTX1 and TRIM2. 

We have compared our results with two independent datasets to investigate reproducibility. We observed that only a small percentage of miRNAs overlapped with the independent datasets. The independent validation results also lack consistency when DMFS and OS signatures were assessed for predicting distant relapse-free survival (DRFS) and OS. Sample scarcity, a different analytical platform and demographic changes may have accounted for inter-study differences. The heterogeneity with in TNBC could have also played an important role in the performance of miRNAs in their prognostic ability. 

These findings suggested that the gene clusters identified during the mRNA profiling can be influenced by the targeting of significant miRNAs. These findings also support anti-correlative relationship of OS and DMFS prognostic miRNAs with oncogenesis/suppression of cancer-related pathways in predicting prognosis. Some of the prognostic miRNAs observed in this study are somehow linked to various cancers, but this is the first time that we have identified TNBC subtype-specific miRNAs associated with OS and DMFS survival. MiRNAs such as hsa-miR-342-3p, hsa-miR-195, hsa-miR-155, and hsa-miR-497 were associated with OS, whereas hsa-miR-29c, hsa-miR-342-3p, hsa-let-7c, hsa-let-7b, and hsa-miR-497 were correlated with DMFS. We have also observed that hsa-miR-630, hsa-miR-195, and hsa-miR-101 have contributed to drug resistance in breast cancers and hsa-miR-497 in pancreatic cancer previously [66]. 

## 4. Material and Methods 

### 4.1. Data and Pre-Processing

miRNA expression profiles detected by Agilent-021827 platform were obtained from GEO (Gene Expression Omnibus) and deposited under the accession number GSE40267. The miRNA expression data consist of 134 breast cancers and eight normal-like samples with extensive patient follow-up and pathological information. Similarly, the mRNA expression profiles of 149 invasive breast cancers from the 172 specimens detected by Illumina human Ref-8 platform were obtained from GEO deposited under the series accession number GSE16987. 

Samples were divided into breast tumour subgroups by carefully examining the receptor status of three known biomarker genes Er, Pr, and Her2. Raw expression datasets were extracted using Bioconductor R ‘GEOquery’ package [67] for both types of data. Quality reports were generated using ‘preprocesscore’ package of Bioconductor R for potential outliers. Expression values were quantile-normalised and log-transformed using default functions of ‘limma’ Bioconductor R package. We performed *lmFit()* function in order to identify differentially expressed miRNAs among by selecting the coefficients of normal versus TNBC, normal versus non-TNBC and TNBC versus non-TNBC groups (Appendix A). miRNAs with *p*-value < 0.05 were considered for downstream analysis. P-values were adjusted for multiple testing using the False Discovery Rate method. We used R base *hclust* function for the calculation of two-dimensional average-linkage hierarchical clustering by building the Spearman rank correlation matrix. Heat maps were draw using ‘gplots’ package of R Bioconductor for manual visualisation. We applied ‘silhouette’ for selecting the optimal clusters for miRNA and mRNA datasets (Appendix A). 

### 4.2. Collection of Predicted, Validated Targets and Calculation of Correlation Index

For our integrated analysis, we applied five widely used tools for target prediction analysis in order to identify possible interactions between dysregulated miRNAs and mRNA: PITA catalog version 2007 [20], microCosm (also known as miRBase Sequence database) [68] and TargetScan catalog version 6.2 [21]. Similarly, we selected two databases for validated targets: TarBase version v.5c [69] and miRecords version April 27, 2013 [70]. An in-house SQLite (http://CRAN.R-project.org/package=RSQLite) database was built to merge, store and process data of all five prediction tools. In the second step, a fast querying searching was applied for the collection of miRNAs targets for downstream analysis. To evaluate the relationship between miRNAs and gene targets we performed correlation analysis using the expression values of each miRNA and their potential target genes. We particularly focused on inversely expressed miRNA–mRNA pairs because of the mechanism of miRNA gene regulation and degradation during the transcription process. Finally, top anti-correlated pairs were selected and further investigated for their functional impact on the clusters of mRNA expression profiles. An interacting network was build using Cytoscape [71] tool visualising the contradicting effects of miRNAs (Figure 3). 

### 4.3. Statistical Analysis

A two-step process was adopted for survival model. Firstly, univariate Cox-regression was performed on the 172 differentially expressed miRNAs for their association with overall survival (OS) and distant metastatic-free survival (DMFS) using ‘survival’ R base package. *p*-values were adjusted for multiple-tests using the Benjamin–Hochberg method. miRNAs were ranked according to their p-value, and those exhibiting *p*-value < 0.05 were considered for multivariate modelling, resulting in an optimal set of miRNAs being derived for both outcome conditions. Censoring occurred at the date of death from any cause (overall survival (OS)), and first evidence of metastases (distant metastatic-free survival (DMFS)) or at the time of 10-year follow up, depending upon the first occurrence of any of them. Different Cox-regression models were applied for the selection of pathological covariates. Histopathological covariates were grouped as Tumour Grading (1 and 2 as low, 3 = high), Node Positivity (positive or negative), Tumour size (continuous variable), % of Tumour Cell Infiltration (low < 50% and high > 50%), Age Group (group A > 50 years, group B < 50 years). We further assessed the quality of each fitted model using the analysis of deviance likelihood test for the selection of co-variate, which could impact the association of prognostic factors with miRNA expression on the outcome prediction (Appendix A). Log-rank and Kaplan–Meier analyses were conducted among the three groups (low, intermediate and high) expressing samples for the calculation difference and drawing of the survival curves using ‘survcomp’ R package. 

### 4.4. Gene Set Enrichment Analysis of Predicted Targets

We performed functional enrichment analysis for each cluster of the mRNA data in order to identify the potential involvement of cellular processes, molecular functions and biological pathways. Two different analysis were performed for gene ontology enrich terms and gene set enriched pathways using ranked lists from publically available databases such as DAVID [72], KEGG [73], Reactome [74] and CGAP [75] to translate the expression profiles of mRNA clusters Figure 1A (See Appendix A for detailed workflow). 

### 4.5. Independent Cohorts for Validation for miRNA Prognostic Signature

We performed “in silico” validation of prognostic miRNAs using two independent datasets published by Bockhorn et al., with 18 TNBC and 28 Non-TNBC [76], and Buffa et al., including 210 Non-TNBC [77] breast tumour samples. The ability of the model to predict outcome was assessed by calculating the AUC of respective ROC curves for the period of 10 years. Higher AUC indicates better performance of the model (AUC = 0.5). 

## 5. Conclusions

In conclusion, we have identified subtype-specific signatures from integrated analyses of miRNAs and mRNAs expression profiles. The results show that the identified miRNAs are prognostic markers for OS and DMFS and directly control the important oncogenic processes and pathways through modulating important cancer driver genes and pathways. The study findings suggested a lack of consistency when DMFS and OS signatures were assessed for predicting the distant relapse-free survival (DRFS) and OS using data from two independent cohorts. Further investigation of tumour progression from normal to tumour subtypes has shown only three miRNAs differentially expressed across the tissues and triple-negatives acquired the highest expression change. The study provides further insight to the complex heterogeneity in triple-negatives and also provides evidence for miRNA as an influencing factor on tumour transcriptional phenotypes at the transcriptome level. The study also shows that the model-based breakdown of system-genomics changes from the dysregulation of epigenetic miRNAs to the transcriptomic pathways. Our results suggest that molecular studies based on miRNAs’ biomarkers can help in the early detection of disease and can used as agents of therapeutic interventions in triple-negatives. 

## Figures and Tables

**Figure 1 cancers-12-00632-f001:**
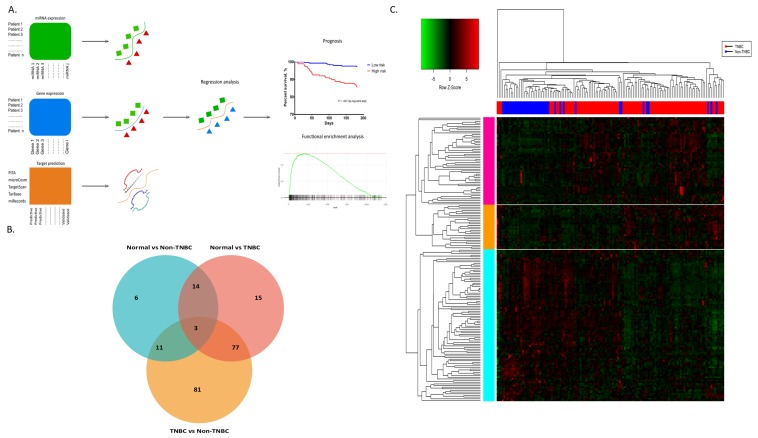
(**A**) Schematic workflow diagram of the integrative analysis of mRNA and miRNA expression profiles. (**B**) Venn diagram of number of differentially expressed miRNAs. Genes in overlapping sets shows differential expression in three comparisons of normal versus triple-negative breast cancer (TNBC), normal versus non-TNBC and TNBC and non-TNBC pairs. (**C**) Heat map of two-dimensional hierarchical clustering of 172 differentially expressed miRNAs among the TNBC versus non-TNBC comparisons. Three major clusters of miRNAs were found and labelled in different colours (See Appendix A).

**Figure 2 cancers-12-00632-f002:**
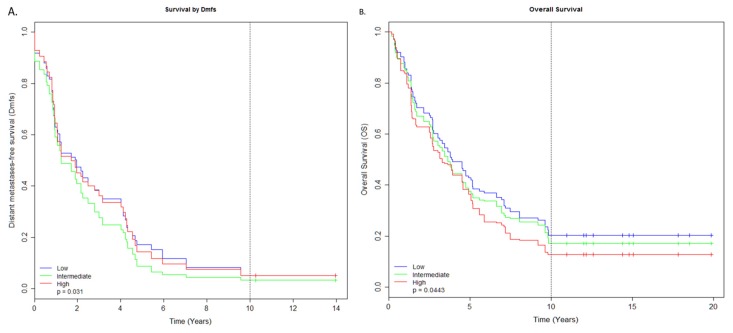
Figure showing the results from log-rank test among the three groups of (low, intermediate and high) expression samples. (**A**) Kaplan–Meier curves showing the difference in three groups with disease metastatic-free survival (DMFS). (**B**) Kaplan–Meier curves showing the difference in three groups with overall survival (OS).

**Figure 3 cancers-12-00632-f003:**
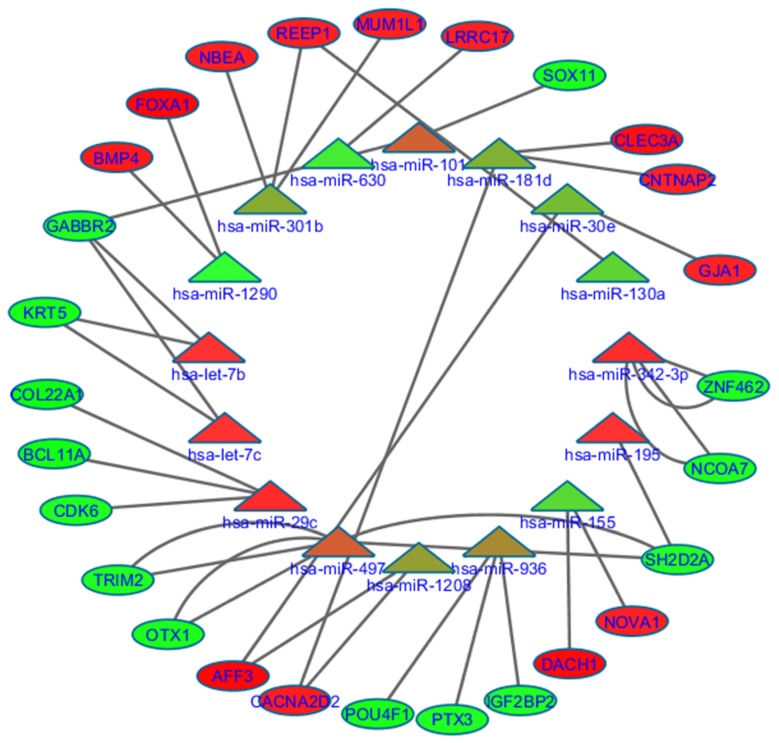
miRNA–mRNA interaction network showing the inverse correlative impact of miRNA to their respective targeting genes from the mRNA expression dataset. The inner track is 16 miRNAs and the outer track is a consensus gene sets of 27 mRNA genes between differentially expressed mRNA and predicted/validated targets from five databases. The miRNAs/genes in green show significant downregulation and red represent the upregulation of targeting genes.

**Table 1 cancers-12-00632-t001:** Table showing the results of cox regression model of the miRNAs significantly associated with distance metastasis–free survival. The regulation column represents the trends of expression compare to non-TNBC samples.

miRNAs	HR	Lower	Higher	*p*-Value	TNBC vs Non-TNBC		
					Regulation	*p*-Value	Type
hsa-miR-29c	0.72	0.52	1	0.0469	Up	1.73E-16	Protective
hsa-miR-342-3p	0.52	0.31	0.89	0.0162	Up	2.99E-12	Protective
hsa-miR-342-5p	0.3	0.1	0.93	0.0356	Up	1.49E-09	Protective
hsa-let-7c	0.63	0.41	0.98	0.0411	Up	4.25E-06	Protective
hsa-miR-19b-1 *	0	0	0.69	0.0374	Down	9.29E-06	Protective
hsa-let-7b	0.5	0.31	0.83	0.0057	Up	9.60E-06	Protective
hsa-miR-1290	1.71	1.2	2.43	0.0022	Down	2.61E-04	Risky
hsa-miR-369-5p	0	0	0.42	0.0262	Up	5.27E-04	Protective
hsa-miR-301b	5.31	1.13	24.96	0.0324	Down	6.60E-04	Risky
hsa-miR-630	1.64	1.17	2.3	0.0029	Down	2.26E-03	Risky
hsa-miR-101	0.58	0.33	1	0.0486	Up	7.98E-03	Protective
hsa-miR-1246	1.53	1.12	2.09	0.0071	Down	1.03E-02	Risky
hsa-miR-181d	0.31	0.1	0.95	0.0382	Down	1.13E-02	Protective
hsa-miR-181c *	0.1	0.01	0.76	0.0244	Down	1.39E-02	Protective
hsa-miR-30e	0.49	0.25	0.98	0.0436	Down	1.63E-02	Protective
hsa-miR-497	0.51	0.29	0.9	0.0193	Up	2.30E-02	Protective
hsa-miR-154	0.05	0	0.58	0.0168	Up	3.28E-02	Protective
hsa-miR-130a	0.5	0.33	0.78	0.0017	Down	4.22E-02	Protective

* indicates miRNA originating from same hairpin structure (pri and pre-miRNA) of the main miRNA.

**Table 2 cancers-12-00632-t002:** Table showing the results of cox regression model of miRNAs significantly associated with overall survival. The regulation column represents how the trends of expression compare to non-TNBC samples.

miRNA	HR	Lower	Higher	*p*-Value	TNBC vs Non-TNBC		
					Regulation	*P*-Value	Type
hsa-miR-342-3p	0.68	0.5	0.92	0.0127	Up	2.99E-12	Protective
hsa-miR-342-5p	0.39	0.2	0.75	0.00415	Up	1.49E-09	Protective
hsa-miR-193b	1.5	1	2.25	0.0487	Up	3.23E-09	Risky
hsa-miR-195	0.76	0.59	0.98	0.0325	Up	1.56E-03	Protective
hsa-miR-155	0.61	0.41	0.91	0.0157	Down	7.44E-03	Protective
hsa-miR-936	5.79	1.04	32.08	0.0442	Up	1.17E-02	Protective
hsa-miR-338-3p	0.43	0.19	0.96	0.0377	Up	1.40E-02	Protective
hsa-miR-1208	376.22	10.32	13709.16	0.00111	Down	1.78E-02	Risky
hsa-miR-497	0.64	0.44	0.94	0.021	Up	2.30E-02	Protective
hsa-miR-146b-5p	0.65	0.45	0.94	0.0212	Down	2.39E-02	Protective

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
