# Peer review of "Model-Based Integration Analysis Revealed Presence of Novel Prognostic miRNA Targets and Important Cancer Driver Genes in Triple-Negative Breast Cancers"

_cancers, 2020, doi:10.3390/cancers12030632_

Round 1

Reviewer 1 Report

The authors performed a model-based integrated analysis of miRNA and mRNA expression profiles on breast cancer on triple-negative to identify subtype-specific signatures involved in oncogenic pathways and their potential role in patient survival outcomes. They used univariate and multivariate Cox analysis to identify 25 miRNAs associated with prognosis of overall survival (OS) and distant metastases-free survival (DMFS) with “risky” and “protective” outcomes. They provide new insights into the potential role of miRNAs in TNBC disease progression through the activation of key oncogenic pathways.

Whilst the writing is clear, the effort is commendable, and the authors present an interesting approach to identify new functional clusters of genes from complex patient datasets, in my opinion, there are some concerns that should be addressed before proceeding with this work.

1. Fig. 1A the workflow needs to be elaborated in more detail in order to maintain the reproducibility of the analysis. Perhaps a supplementary document or a GitHub page would be appropriate.

2. Three major clusters identified in Fig 1C need to be included as tables with annotation in a supplementary table

3. The authors performed functional enrichment analysis on two large clusters derived from 386 mRNAs. It seems to me this is the Hypergeometric test. It needs to be clarified in the method section.

4. Also, the Gene ontology enrichment analysis in the Methods section needs to be clarified. The heading indicates GO enrichment analysis but in the text, the author stated: "available databases such as DAVID [73], KEGG [74], Reactome [75] and CGAP [76] to translate the expression profiles of mRNAs clusters. ". While DAVID can be used for GO analysis, KEGG and Reactome are canonical pathways databases and should not be confused with Gene Ontology. The authors should amend the text and refrain from making unjustified conclusions.

Author Response

The authors performed a model-based integrated analysis of miRNA and mRNA expression profiles on breast cancer on triple-negative to identify subtype-specific signatures involved in oncogenic pathways and their potential role in patient survival outcomes. They used univariate and multivariate Cox analysis to identify 25 miRNAs associated with prognosis of overall survival (OS) and distant metastases-free survival (DMFS) with “risky” and “protective” outcomes. They provide new insights into the potential role of miRNAs in TNBC disease progression through the activation of key oncogenic pathways.

Comments/Questions:

  1. Whilst the writing is clear, the effort is commendable, and the authors present an interesting approach to identify new functional clusters of genes from complex patient datasets, in my opinion, there are some concerns that should be addressed before proceeding with this work. 1A the workflow needs to be elaborated in more detail in order to maintain the reproducibility of the analysis. Perhaps a supplementary document or a GitHub page would be appropriate.

REPLY: We agree with reviewer 1 that inclusion of detailed workflow highlighting each preprocessing and data analysis steps in a separate supplementary file to maintain reproducibility of the work. Figure 1-C and supplementary information Figure S-1 is included for the data analysis workflow along with the detail of each step such as data acquisition, preprocessing, analysis and results.

  1. Three major clusters identified in Fig 1-C need to be included as tables with annotation in a supplementary table.

REPLY: We have added information of the three cluster in the form of a table to address this comment (Supplementary information Table S-1).

  1. The authors performed functional enrichment analysis on two large clusters derived from 386 mRNAs. It seems to me this is the Hypergeometric test. It needs to be clarified in the method section.

REPLY: We added more details to the method section to address this comment (see supplementary methods section 4 on functional enrichment analysis).

  1. Also, the Gene ontology enrichment analysis in the Methods section needs to be clarified. The heading indicates GO enrichment analysis but in the text, the author stated: "available databases such as DAVID [73], KEGG [74], Reactome [75] and CGAP [76] to translate the expression profiles of mRNAs clusters. ". While DAVID can be used for GO analysis, KEGG and Reactome are canonical pathways databases and should not be confused with Gene Ontology. The authors should amend the text and refrain from making unjustified conclusions.

REPLY: We agree with reviewer 1 that the heading was misleading and now has been corrected (main text material and methods section). To clarify this point, we have performed two separate analysis of gene ontology enrichment analysis and pathway analysis from gene sets generated from different database such as KEGG, MSigDB, Reactome and CLLG. Two separate tables of top GO term and regulatory pathways have been added to the supplementary file to avoid any confusion and the changes have been added to the method

Reviewer 2 Report

The identification of cancer-specific miRNA  is a novel strategy in the disease diagnosis. However, It is been widely accepted that the use of miRNA as a disease-specific biomarker is not well defined and result from independent studies often contradict. However, hunting of unique disease-specific miRNA targets can not be ignored too. The computational methods for data processing steps have their own challenges including normalization miRNA invariant adjustment etc. Selecting the right tools for miRNA data processing and strict standardization of process is critical in these studies.  Differentially expressed single or two miRNA often unrelated to disease and leads to false positive or false negative prognosis, therefore the use of a panel of miRNA might give more valuable information compared to single or two miRNA markers. The authors comprehensively elucidated data for miRNA to be used as prognostic markers. Since several miRNA expressions depend on the specific cell cycle, did the author considered cell cycle phase-specific expression of miRNA in this study? Predominantly, overexpressed miRNA mask the low expression miRNA during amplification, did the author corrected the data to avoid these errors? It would be more informative if the principal component (PC%) analysis of the top 10 miRNA expression included and compare normal vs TNBC and TNBC vs non-TNBC.  

There is a need to describe more clearly the choice of miRNA expression normalization in the method section?

Please correct: page 9; last paragraph (1. 0-miRNAs prognostic signature associated with OS), 

Figure 1; It is suggested that top overexpressed 50 miRNA should be plotted into a separate heat map and label the axis properly and readable. Do write a brief methodology used for data processing under footnotes.

Author Response

Comments/Questions:

  1. The identification of cancer-specific miRNA is a novel strategy in the disease diagnosis. However, It is been widely accepted that the use of miRNA as a disease-specific biomarker is not well defined and result from independent studies often contradict. However, hunting of unique disease-specific miRNA targets cannot be ignored too. The computational methods for data processing steps have their own challenges including normalization miRNA invariant adjustment etc. Selecting the right tools for miRNA data processing and strict standardization of process is critical in these studies.  

REPLY: We agree to the reviews comment about microRNA normalisation is an essential step in the pre-processing stage. Most of the normalisation methods were developed and now currently being applied were in the context of mRNA expression arrays such as median and quantile normalisation methods. These methods rely on the information that only small proportion of the fixed probes are differentially expressed as compared nearly every probe of microRNAs arrays. The biology and the behaviour of microRNAs is very different to the mRNA at every level, starting from huge probe numbers difference on the arrays to likely higher differential expression in the normal and tumour tissues. The choice of normalisation is also dependent on the type of colour and platform of individual arrays. For this study, Agilent arrays were used for the extraction of microRNA for this data set and the most widely used data normalisation method is ‘limma’ R package ‘quantile normalisation’ for this particular platform. Multiple studies (Hua et al 2008; Pradervand et al 2009; Rao et al 2008; Bargaje et al 2010) have evaluated the performance of quantile normalisation method and have confirmed as one of the most robust methods for microRNA preprocessing steps. According to manufacturer protocol, a simple linear scaling of is sufficient for this particular array and its application (page 52 (https://www.agilent.com/cs/library/usermanuals/public/G4170-90011_miRNA_Protocol_3.1.pdf)). Quality control box plots will be included in the supplementary file showing differences of before and after normalisation.

  1. Differentially expressed single or two miRNA often unrelated to disease and leads to false positive or false negative prognosis, therefore the use of a panel of miRNA might give more valuable information compared to single or two miRNA markers.

REPLY:  We agree to reviewer’2 comment about the panel of miRNA providing more robust information as compared the single or two miRNA. Our multivariate analysis shows that the panel of 18 miRNAs and 10 miRNAs are independent markers and clearly separate the patients with low and intermediate and high expression. Supplementary Table S6-7 confirms the significance of the prognostic markers. We have revised the method and discussion to more accurately represent these findings.   

  1. The authors comprehensively elucidated data for miRNA to be used as prognostic markers. Since several miRNA expressions depend on the specific cell cycle, did the author considered cell cycle phase-specific expression of miRNA in this study?

REPLY: The reviewer 2 asked a very pertinent questions related to the biology of microRNA but in the scope of study is model based integrated analysis of literature generated from triple-negative breast tissues. Further analysis of cDNA sequences of microRNA groups can be explored to check the S-phase library or it can be investigated during the cDNA hybridisation stage.     

  1. Predominantly, overexpressed miRNA mask the low expression miRNA during amplification, did the author corrected the data to avoid these errors? It would be more informative if the principal component (PCA) analysis of the top 10 miRNA expression included and compare normal vs TNBC and TNBC vs non-TNBC.

REPLY: We have performed PCA analysis of the top 10 regulatory miRNAs from the comparison of normal versus triple negative, normal versus non-triple negative and triple negative versus non-triple negative. A figure of PCA plots is included in the supplementary information (Figure S-6).

  1. There is a need to describe more clearly the choice of miRNA expression normalization in the method section?

REPLY:  We agree to the reviewer’s to comments and have revised the description of normalisation method used in the material and methods section also supplementary methods heading (mRNA and microRNA data extraction and preprocessing) as suggested. A part it already been described in the Comments/Question 1.

  1. Please correct: page 9; last paragraph (1. 0-miRNAs prognostic signature associated with OS), 

REPLY: Correction has been made as suggested and highlighted in yellow.

  1. Figure 1; It is suggested that top overexpressed 50 miRNA should be plotted into a separate heat map and label the axis properly and readable. Do write a brief methodology used for data processing under footnotes.

REPLY: We agree to the statement of inclusion of anther heat map. We have plots heatmap of the three cluster separately in the supplementary information (Figure S3-5) to address this comment.

Round 2

Reviewer 2 Report

The author has edited and improved the manuscript. I didn't find major objections in the current format of the manuscript and can be accepted for publication.